# Ionic covalent organic framework based electrolyte for fast-response ultra-low voltage electrochemical actuators

Fei Yu [1], Jing-Hao Ciou[1], Shaohua Chen[1], Wei Church Poh[1], Jian Chen[1], Juntong Chen[1], Kongcharoen Haruethai[1], Jian Lv [1,2], Dace Gao[1] & Pooi See Lee [1,2 ✉]

Electrically activated soft actuators with large deformability are important for soft robotics but enhancing durability and efficiency of electrochemical actuators is challenging. Herein, we demonstrate that the actuation performance of an ionic two-dimensional covalent-organic framework based electrochemical actuator is improved through the ordered pore structure of opening up efficient ion transport routes. Specifically, the actuator shows a large peak to peak displacement (9.3 mm, ±0.5 V, 1 Hz), a fast-response time to reach equilibrium-bending (~1 s), a correspondingly high bending strain difference (0.38%), a broad response frequency (0.1–20 Hz) and excellent durability (>99%) after 23,000 cycles. The present study ascertains the functionality of soft electrolyte as bionic artificial actuators while providing ideas for expanding the limits in applications for robots.

[1] School of Materials Science and Engineering, Nanyang Technological University, Singapore 639798, Singapore. [2] Singapore-HUJ Alliance for Research and Enterprise (SHARE), Nanomaterials for Energy and Water Nexus (NEW), Campus for Research Excellence and Technological Enterprise (CREATE), Singapore 639798, Singapore. ✉email: pslee@ntu.edu.sg

Bionic artificial muscles that exhibit reversible motion responses upon external stimuli (heat, pH, humidity, salt, solvent, electric or magnetic field, and light) have made remarkable progress over the past decades[1–13]. Because of their flexibility[14,15], low cost, low pollution, no noise, lightweight, low activation voltage, versatile deformations, and rapid response[16], electroactive polymer (EAP) actuators have emerged as versatile materials for high-efficiency micro- and macro-artificial muscles and soft robotic systems[17,18], biomedical devices[19], and biomimetic flying insects[20]. In particular, ionic-polymer metal composites (IPMCs) actuators are positioned as promising candidates due to their decent bending actuation at ultralow voltages and are among the most promising EAP materials for artificial muscle constructed with ionic-conductive electrolytes, mobile molten ionic salts, and sandwiches with metallic conductors[21–25]. The deformation of the IPMC actuator is based on the volume or pressure gradient of the actuator, reversible ion insertion, and de-intercalated under the applied electrical stimulation[26]. Over the past decades, various types of actuators have been widely explored[27–30]. However, the long-term challenge remains in designing and manufacturing actuators with high electromechanical conversion efficiency, stronger mechanical output force, fast response, large stress/strain density, and operational durability in the air, as the present IPMC actuation performances cannot meet the demands for practical applications[31–33].

Ionic covalent organic frameworks (iCOFs), as one of the rapidly developing subclasses of crystalline materials, have been developed because of their regular porosity architectures and high ionic conductivities[34–38]. The distinctive structural and chemical characteristics and high Brunauer–Emmett–Teller (BET) surface area offer appealing benefits for increased ion transport via the frameworks. These features can usually be used in ion-conducting[34,39–41], proton conduction[35,36], iodine capture[42], catalysis[43], fuel cell[44], electro/photochromic[45–47], and gas separation[48]. The tubular one-dimensional (1D) hole, created between the two-dimensional (2D) COF layers, may provide effective ion-transport pathways that are associated with the impact of the confinement and relative ions and pore size. To the best of our knowledge, the potential of high-performance two-dimensional ionic COF-based electrolytes in electrochemical actuators has largely remained unexplored.

In this work, a two-dimensional (2D) ionic COF (COF-DT-$SO_3Na$) with 1,3,5-tris(4-aminophenyl)benzene (TAB) and 2,5-dihydroxyterephthalaldehyde (DHA) building blocks is prepared (Fig. 1) and employed as an active element in electrochemical soft actuators. The COF-DT-$SO_3Na$-based electrolyte actuator has a high actuation displacement (9.3 mm, 1 Hz), a bending-strain difference of 0.38% (1 Hz), a strong resonance displacement (~12.1 mm, 8 Hz), a rapid time to reach equilibrium-bending motion (~1 s), a wide-frequency (0.1–20 Hz) response, and long-term durability (>23,000 cycles) under continuous electrical stimuli in the air with negligible degradation in actuation performance under ±0.5 V at a frequency of 0.1 Hz. The increased actuation performance is due to both the oriented porous structure and efficient transfer routes for rapid ion transport while facilitating significantly high diffusion rates[34,39–41]. In addition, the excellent voltage-driven deformation of COF-DT-$SO_3Na$ actuators can mimic biological motions, such as flowering, tendril curling, and butterfly high-frequency flapping, which makes COF-DT-$SO_3Na$ actuators as a promising candidate for future biomimetic robots or soft biomedical end effectors.

## Results

### COF design and synthesis.
The building blocks of TAB and DHA were synthesized based on the previous methods with slight modifications (Supplementary Methods)[49]. The 2D COF (COF-DT) was synthesized as follows: TAB and DHA were first dissolved in the solvent of n-butanol:1,2-dichlorobenzene:acetic acid (6 M) (v/v/v = 5:5:1), then the mixture was ultrasonicated for 30 minutes to generate a red slurry, followed by freezing in liquid nitrogen at 77 K, degassing through four freeze–pump–thaw cycles, and sealed in a Schlenk flask. The mixture was heated at 120 °C for 96 h and yielded red precipitates of COF-DT. Then, the product reacted with 1,3-propanesultone at 110 °C in toluene for 6 h, allowing the sulfonate ester to react with the OH functional group of COF-DT. This results in the alkyl chain with the terminal sulfonic acid functional group fixed inside the channel, giving dark red precipitates of COF-DT-$SO_3H$. In the end, the COF-DT-$SO_3H$ reacted with NaOH (1 M) at room temperature for ion-exchange reaction with the sulfonic acid functional groups and yielded the final product as COF-DT-$SO_3Na$ (Fig. 2).

### Structure characterization.
The as-synthesized COF-DT powder X-ray diffraction (PXRD) pattern displays several discrete Bragg reflections with $2\theta$ roughly at 2.8, 4.9, 5.6, and 7.4°, which are compatible with (1 0 0), (1 1 0), (2 0 0), and (1 2 0) faces in the modeled structure. In addition, Pawley refinement was utilized to uncover the $P6/m$ space group and a unit cell with $a = 38.19$, $b = 38.19$, $c = 3.48$ Å, and $\gamma = 120°$ with $R_{wp} = 4.33\%$ and $R_p = 3.34\%$. According to their difference, there is a good agreement between the experimental data and the refined pattern (Fig. 3a). The diffraction patterns of COF-DT-$SO_3H$ and COF-DT-$SO_3Na$ are in good match with the diffraction pattern of COF-DT (Fig. 3b), indicating that the framework remains stable after post-synthetic modification. Transmission electron microscope (TEM) samples were prepared by immersing COF-DT powder in ethanol and sonicating it at a power of 100 W and a frequency of 37 kHz for 60 minutes. The TEM image of the sample (Fig. 3c) clearly shows parallel lines with an adjacent distance of ~3.4 Å, which is attributed to the stacking structure around the c-axis of the COF-DT. Furthermore, the attenuated total reflection Fourier transform-infrared (ATR FT-IR) spectra of COF-DT-$SO_3H$ and COF-DT-$SO_3Na$ show two additional peaks at 1034 and 1042 cm$^{-1}$, corresponding to S = O stretching vibration, which verified the effective integration of sulfonic groups into COF-DT (Supplementary Fig. 3)[49]. The energy-dispersive X-ray (EDX) spectroscopy showed homogeneously dispersed emissions from C, N, and O for COF-DT, from C, N, O, and S for COF-DT-$SO_3H$, and from C, N, O, S, and Na for COF-DT-$SO_3Na$ (Supplementary Fig. 4).

Brunauer–Emmett–Teller (BET) study indicates that COF-DT has a reversible adsorption isotherm of $N_2$ at 77 K, resulting in a BET surface area of 1287 m$^2$ g$^{-1}$ with a pore diameter of ~3.2 nm by using non-local density functional theory (NLDFT) modeling. For the post-synthesized compounds of COF-DT-$SO_3H$ and COF-DT-$SO_3Na$, the BET surface areas are 470 and 381 m$^2$ g$^{-1}$, with the pore diameters of ~2.5 and 2.4 nm, respectively (Fig. 3d and Supplementary Fig. 5). These results demonstrate that the effects on the crystallinity and pore structure of the original material are negligible by post-synthetic modification. In addition, these nanopores can provide necessary space and act as nanoreactors to transfer oppositely charged ions without stress during the switching of AC input signals in the presence of available electrolytes.

Due to the large number of sulfonated groups neatly arranged in one-dimensional nanoporous channels, COF-DT-$SO_3Na$ is expected to be a highly ionic conductive material[37,38]. To enhance ion migration, 10 μL of EC/DMC (v/v = 1:1) mixed solvent was added to COF-DT-$SO_3Na$ as a plasticizer. Temperature-dependent (20–80 °C) electrochemical impedance spectroscopy (EIS) of COF-DT-$SO_3Na$ revealed a superior conduction

**Fig. 1 Synthesis of ionic COF.** Synthetic route of ionic COF-DT-SO₃Na.

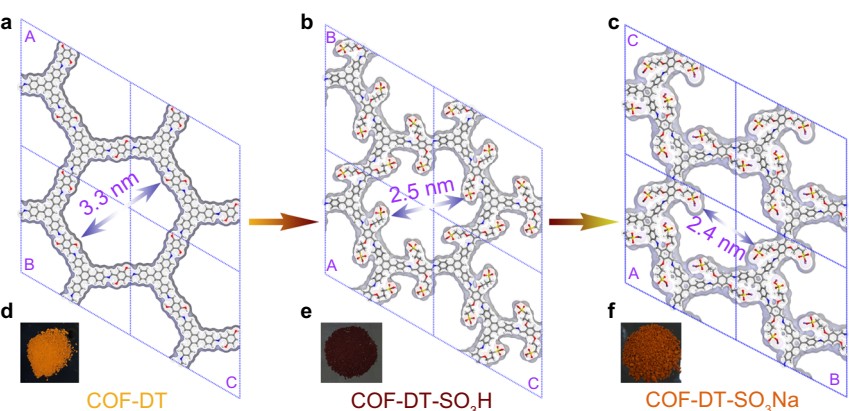

**Fig. 2 Structural aspects of COF.** Simulated structure of **a** COF-DT, **b** COF-DT-SO₃H, and **c** COF-DT-SO₃Na (Colors: blue—N, gray—C, red—O, yellow—S, pink—Na, white—H) with the pore sizes of 3.3, 2.5, and 2.4 nm, respectively. The powder-sample images under visible light of **d** COF-DT, **e** COF-DT-SO₃H, and **f** COF-DT-SO₃Na.

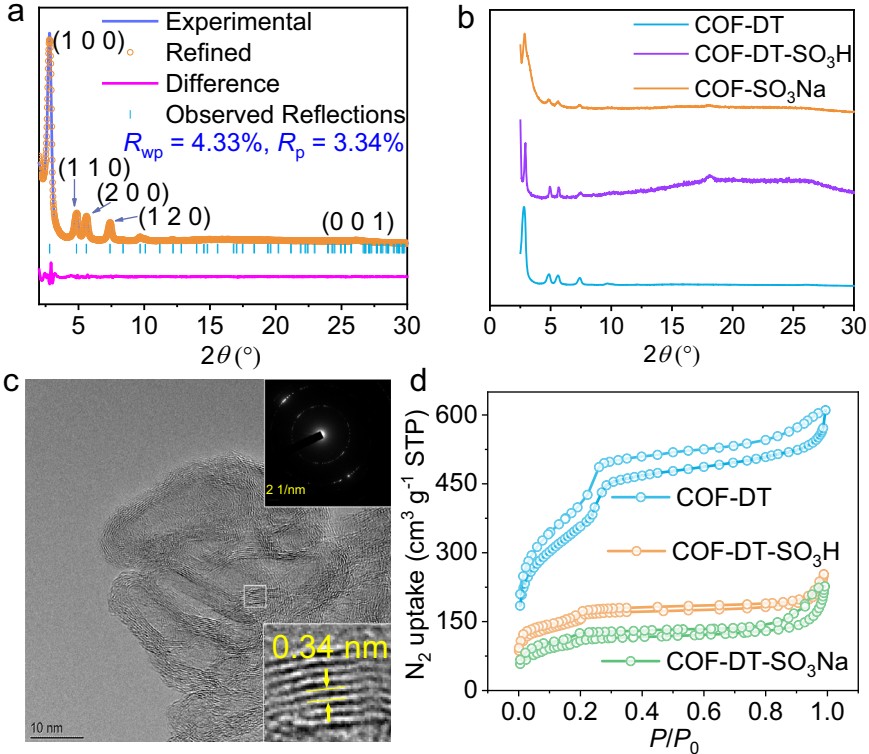

**Fig. 3 Characterizations of COF-DT, COF-DT-SO$_3$H, and COF-DT-SO$_3$Na. a** The experimental powder X-ray diffraction of COF-DT (blue line), the Pawley refined plot (yellow circle), and the difference between experimental and refined plot (purple line). **b** Comparison of experimental powder X-ray diffraction pattern of COF-DT (cyan line), COF-DT-SO$_3$H (green line), and COF-DT-SO$_3$Na (red line). **c** HR-TEM image of COF-DT. **d** Porous properties: N$_2$ sorption isotherm of COF-DT (cyan circle line), COF-DT-SO$_3$H (orange circle line), and COF-DT-SO$_3$Na (green circle line).

performance of the ionic COF pellet. The conductivities (Supplementary Equation 1) were calculated as 13.5, 16.5, 19.4, 23.8, 31.6, 38.4, and 45.2 mS cm$^{-1}$ at 293, 303, 313, 323, 333, 343, and 353 K, respectively (Fig. 4a). Arrhenius fitting result shows that the activation energy ($E_a$) of COF-DT-SO$_3$Na is 0.18 eV (Fig. 4b). To evaluate the potential electronic conductivity of COF-DT-SO$_3$Na, the impedance of the geometrically equivalent COF-DT and COF-DT-SO$_3$H without adding the EC/DMC mixed solvent (25 °C, 30% RH) was also examined. The results reveal that both materials were electric insulators (Supplementary Figs. 6 and 7).

**Actuation performance**. To explore the functional characteristics of COF-DT-SO$_3$Na electrolyte in practical applications, a mixture of COF-DT-SO$_3$Na and EC/DMC/NMP ($v/v/v = 1:1:1$) ionic layer was used as a quasi-solid electrolyte between PEDOT:PSS electrodes (Fig. 4c). The cross-sectional morphology of the assembled soft actuator shows an interlayer adhesion between the PEDOT:PSS electrodes and the COF-DT-SO$_3$Na electrolyte, which promotes ion diffusion and significantly reduces ion-transport resistance (Fig. 4c). The CV curve of the device was measured under ±0.5 V at a scan rate from 100 to 1000 mV s$^{-1}$ (Fig. 4d). The non-Faradaic process that occurs at the electrode/electrolyte interface is primarily responsible for the properties of the electric double-layer capacitor. The reversible CV curve gives an area capacity of 125.2 mF cm$^{-2}$ at scan rate of 100 mV s$^{-1}$. The actuation performance of a COF-DT-SO$_3$Na soft actuator with a strip shape (width 4 mm, length 20 mm, and thickness 60 μm) was studied under a square-wave alternating-current (AC) voltage of ±0.5 V at 0.1 Hz, showing that the peak-to-peak displacement ($\delta$) reaches a maximum value of 9.6 mm. In addition, due to the expansion and contraction of the PEDOT:PSS

electrode during the COF-DT-SO$_3$Na ion-intercalation process, reversible deformation was observed in both directions (Fig. 4e and Supplementary Movie 1).

The displacements ($\delta$) shown as a function of frequencies under 0.1–20 Hz at ±0.5 V were also evaluated (Fig. 5a). The actuator reaches the highest displacement (~9.6 mm) at the frequency of 0.1 Hz. From 1 Hz to 0.1 Hz, the maximum displacements of the actuator are almost the same (~9.3 mm, 1 Hz), indicating that the actuator can quickly (~1 s) reach the equilibrium-bending motion (Supplementary Fig. 8). Such fast equilibrium time can be attributed to the nanopores in the COF-DT-SO$_3$Na to which they provide the necessary space to transfer the oppositely charged ions without pressure during the switching process of the AC input signal. The peak-to-peak displacement decreases gradually with increasing frequency at the range of 1–5 Hz. Furthermore, when the frequency reaches an appropriate value (~8 Hz), the soft actuator will resonate strongly, and the swing amplitude will jump to the maximum displacement (~12.1 mm). Moreover, in the range of 9–20 Hz, as the operating frequency increases, the displacement gradually decreases, while above 20 Hz, the actuation performance almost disappears (Fig. 5a and Supplementary Movie 1).

Such excellent actuation performance encourages us to further explore the COF-DT-SO$_3$Na actuator in the air at the lower voltage. The maximum displacement becomes 6.7 and 2.7 mm under voltages of ±0.3 and ±0.1 V at 0.1 Hz, respectively. The resonance behavior can be also determined near the frequency of 8 Hz, resulting in the displacement of 5.8 and 2.4 mm at ±0.3 and ±0.1 V, respectively. Moreover, in the range of 9–20 Hz, as the operating frequency increases, the displacement gradually decreases, while above 20 Hz, the actuation performance almost disappears (Fig. 5b).

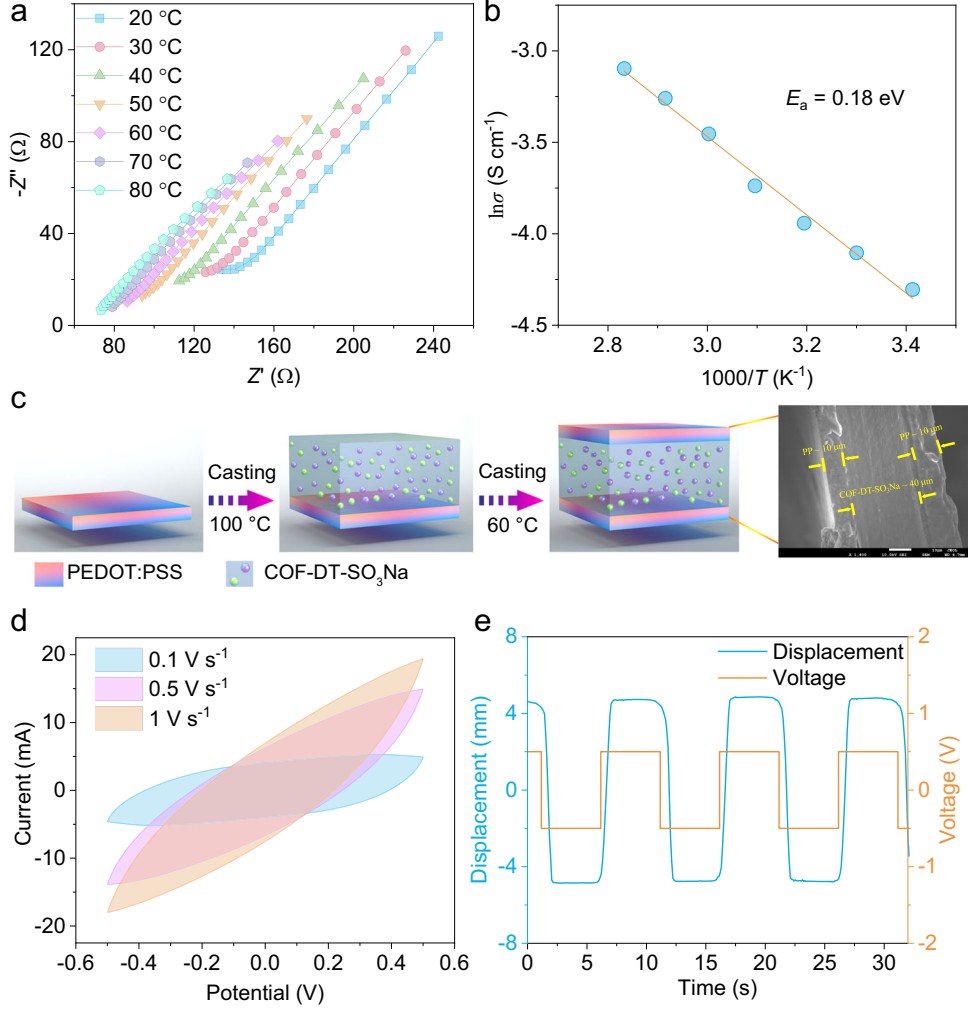

**Fig. 4 Characterizations of COF-DT-SO₃Na-based actuator. a** EIS plot of COF-DT-SO₃Na. In all, 20 °C (blue quadrilateral line), 30 °C (red circle line), 40 °C (green upward triangle line), 50 °C (orange downward triangle line), 60 °C (pink quadrilateral line), 70 °C (cerulean-blue hexagon line), 80 °C (cyan pentagon line). **b** Arrhenius fitting (orange line) of ln (conductivity) versus scaled inverse temperature, experimental data (blue circle). **c** The process of fabricating DT-COF-SO₃Na-based soft actuators. **d** The CV curves of the COF-DT-SO₃Na based soft actuator at various scan rates, 0.1 V s⁻¹ (blue), 0.5 V s⁻¹ (red), and 1 V s⁻¹ (orange). **e** The displacement (blue line) of COF-DT-SO₃Na-based soft actuator under an AC square-wave voltage (orange line) of ±0.5 V at 0.1 Hz.

The actuator produces 0.39%, 0.30%, and 0.14% bending-strain differences (%) (Supplementary Equation 2) under ±0.5, ±0.3, and ±0.1 V at 0.1 Hz, respectively (Fig. 5c). Specifically, the COF-DT-SO₃Na actuator maintains a relatively large displacement of 0.4 mm and a bending-strain difference of 0.016% at 20 Hz. As a comparison, the Nafion-based actuator has no actuation performance at the frequency above 2 Hz and has a slower equilibrium time (>10 s) (Supplementary Fig. 9). In addition, one of the important aspects of the design of artificial bioinspired actuation systems is the blocking force. Under an alternating-current input voltage of 0.5 V at 0.1 Hz, the blocking force generated 1.2 mN at the tip of the actuator. In addition, the long-term stability of the actuator was also evaluated under ±0.5 V at 0.1 Hz (Fig. 5d). The actuator has negligible degradation after more than 23,000 cycles under continuous electric stimuli. A comparison of the bending performance of the state-of-the-art ionic soft actuators with that of COF-DT-SO₃Na actuator is listed in Supplementary Table 4. It demonstrates that COF-DT-SO₃Na actuator has competitive performance to other materials under ±0.5 V, 1 Hz. A weak butterfly robot was constructed with COF-DT-SO₃Na

actuators as the artificial muscles for wing movement to illustrate the actuator soft robotic capability. The butterfly was adhering to a glass rod and connected to the power supply. The wings of the butterfly soft robot started flapping upon applying a ± 0.5 V under 4- and 5-Hz input electrical stimulus (Supplementary Movie 2). These results confirm that highly oriented and integrated ionic COF-DT-SO₃Na electrolyte provides efficient transfer pathways endowing the electric stimulus with high ionic conductivity, and facilitates the rapid electrolyte migration between the pores, thus giving rise to its fast-responsive actuation nature and long-term stability.

## Discussion

We developed an ionic COF-DT-SO₃Na electrolyte with a highly oriented porous structure for electrochemical soft actuators operating under ultralow electric stimulus (≤ ±0.5 V). The COF-DT-SO₃Na-based ionic soft actuator demonstrates high actuation performance, with large actuation displacement (~9.6 mm), a large strain difference of 0.39%, a fast-attain equilibrium-bending motion (~1 s), a broadband frequency response of 0.1–20 Hz, and

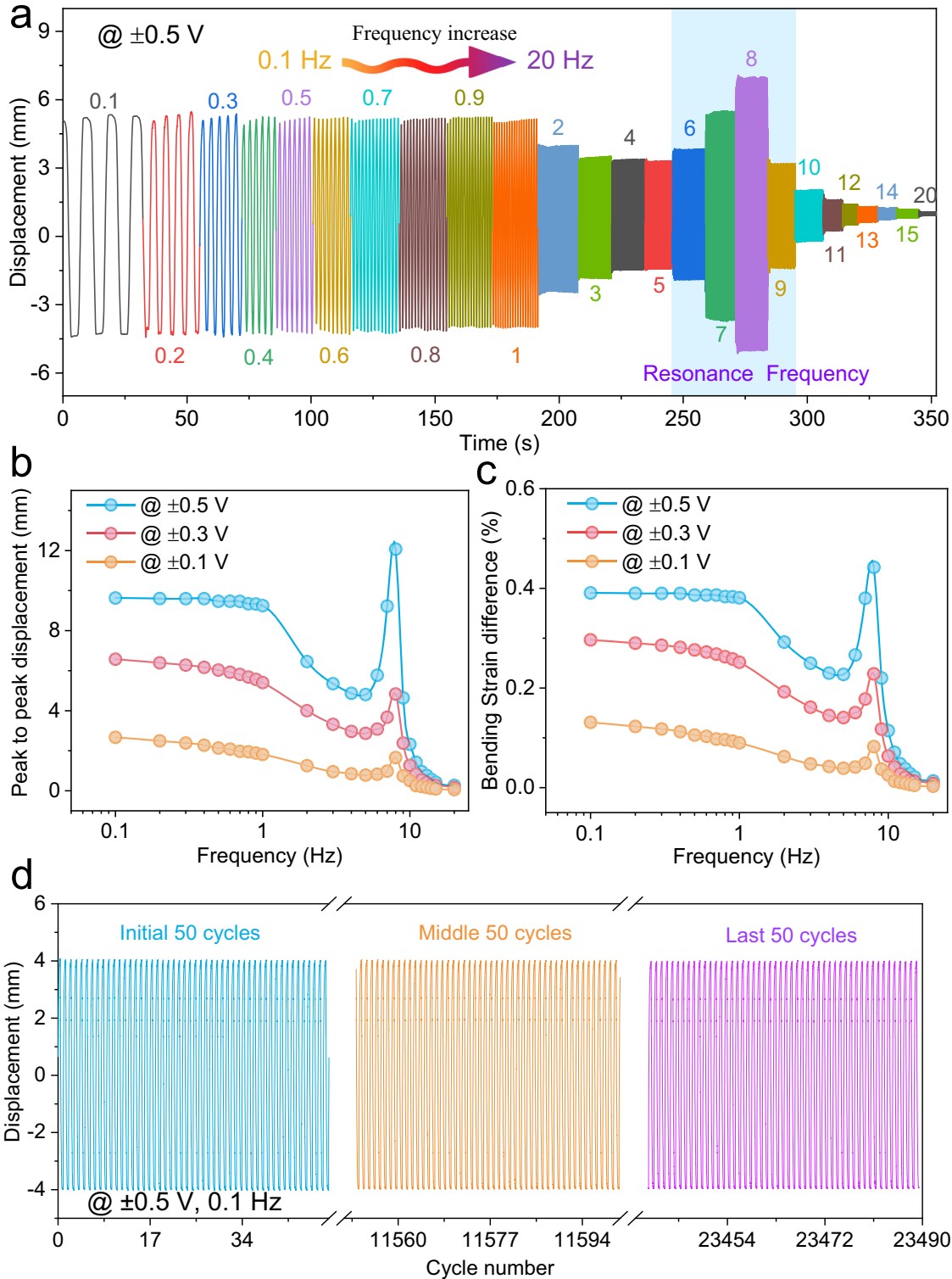

**Fig. 5 The actuation performance of the COF-DT-SO₃Na-based actuator. a** The peak to peak displacement under the frequency of 0.1–20 Hz at ±0.5 V.
**b** Peak-to-peak displacement under the frequency of 0.1–20 Hz at ±0.5 V (blue circle line), ±0.3 V (red circle line), ±0.1 V (orange circle line), respectively.
**c** Bending-strain difference (%) under the frequency of 0.1–20 Hz at ±0.5 V (blue circle line), ±0.3 V (red circle line), and ±0.1 V (orange circle line), respectively. **d** Long-term stability of the soft actuator under ±0.5 V at 0.1 Hz in the air, initial 50 cycles (blue line), middle 50 cycles (orange line), and last 50 cycles (purple line).

long-term durability in the air (>23,000 cycles), thanks to its ordered pores (~2.4 nm), large specific surface area (381 m² g⁻¹), and high ionic conductivity (13.5 mS cm⁻¹). The hierarchical porosity and distinctive COF-DT-SO₃Na electrolyte structure offer efficient transmission and extensive diffusion channels to facilitate mass transportation and ion intercalation/de-intercalation of solid electrical elements in electrodes. We thereby illustrate utilizing a 2D ionically conducting COF-DT-SO₃Na design,

with excellent carrier accessibility that provides excellent actuation performance, further encouraging the development of such materials in the application of actuation. We anticipate that the method described in this study will open up possibilities for the efficient development of high-performance actuator materials by providing a basis for multifunctional actuators.

## Methods

**Synthesis of COF-DT**. About 1.2 mL of a 5:5:1 ($v/v/v$) solution of 1,2-dichlorobenzene/$n$-butylalcohol/6 M aqueous acetic acid was added to a combination of 2,5-dimethoxyterephthalaldehyde (DHA, 24.9 mg, 0.15 mmol) and 1,3,5-tris(4-aminophenyl)benzene (TAB, 35.1 mg, 0.1 mmol) in a Schlenk tube (10 mL). After 30 min of sonication, it was flash-frozen in liquid $N_2$ at 77 K, evacuated, and sealed under vacuum. The mixture was heated at 120 °C for 96 h, yielding an orange precipitate that was separated by filtering and washed with anhydrous tetrahydrofuran for 7 days using Soxhlet extraction. Yielding denoted as COF-DT. Yield: 48.1 mg (80%).

**Synthesis of COF-DT-SO₃H**. COF-DT (500 mg) was evacuated at 100 °C for 6 h, then, cooled to room temperature, and added a mixture of 1,3-propanesultone (0.5 mL) in 30 mL of dry toluene. The resulting mixture was reflux for another 6 h under $N_2$ atmosphere. After being cooled down, the solid was filtered, washed thoroughly with toluene, and dried at 60 °C overnight under vacuum drying oven, yielding a red powder that was denoted as COF-DT-SO₃H.

**Synthesis of COF-DT-SO₃Na**. Methanol (20 mL) was added to a combination of COF-DT-SO₃H (200 mg) and 1 M NaOH (1 mL) in a 50 mL Schlenk flask under $N_2$. COF-DT-SO₃Na was obtained after stirring at room temperature for 6 hours, filtering, and washing with methanol, and drying under vacuum at 50 °C.

**Fabrication of the COF-DT-SO₃Na-based electrochemical actuators**. About 5 wt% of DMSO was added into the aqueous PEDOT:PSS commercial solution. Layer-by-layer sandwiched PEDOT:PSS–COF-DT-SO₃Na–PEDOT:PSS membrane was fabricated by the drop-casting. Specifically, the PEDOT:PSS solution was casted on a glass slide uniformly under 60 °C for 30 min. Then COF-DT-SO₃Na and EC:DMC:NMP ($v/v/v = 1:1:1$) solution (10 mg mL⁻¹) was dropped on the precast PEDOT:PSS membrane and dried at 100 °C for 60 min (Supplementary Fig. 10). The PEDOT:PSS solution was dropped on the PEDOT:PSS–COF-DT-SO₃Na uniformly under 60 °C for 30 min to get a composite actuator.

**Structure characterization**. Powder X-ray diffraction (PXRD) patterns were conducted on a PANalytical X'Pert Pro MPD diffractometer using Cu Kα radiation ($\lambda = 1.5406$ Å) and operating at 40 kV and 40 mA between 2 and 30° ($2\theta$). TEM was performed with a JEM-2100 (JEOL Ltd., Japan) with an accelerating voltage of 200 kV.

## Data availability

The data that support the findings of this study are available within the article and Supplementary Information files, or available from the corresponding authors on request. Source data are provided with this paper.

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

## Acknowledgements

We acknowledge financial support from Singapore Ministry of Education Tier 1, RG63/20 (2020-T1-001-165) and the National Research Foundation, Prime Minister's Office, Singapore, under its Campus for Research Excellence and Technological Enterprise (CREATE) program.

## Author contributions

P.S.L. supervised the project. P.S.L. and F.Y. conceived and designed the project. F.Y., J.H.C., S.H.C., W.C.P., J.C., J.T.C., K.H., J.L., and D.G. synthesized and characterized the materials. F.Y. conceived and built the device and contributed to the actuator device. F.Y. and P.S.L. analyzed the data and wrote the paper. All authors discussed the results and contributed to the paper.

## Competing interests

The authors declare no competing interests.
