## [Peer Review File · Nature Communications]

Ionic Covalent Organic Framework based Electrolyte for Fast-Response Ultra-Low Voltage Electrochemical ActuatorsREVIEWER COMMENTS

Reviewer #1 (Remarks to the Author):

This paper reports a use of COF as electrolyte in ionic actuators. The paper was well written and includes many interesting results. However, although authors claimed at the end of the second paragraph in the introduction that: "As far as we know, the potential of high-performance two-dimensional ionic COF-based electrolytes in artificial actuators has largely remained unexplored", the research presented here has many critical and fundamental issues as pointed out following:

1, The material is not totally new. Many papers 1-4 reported about the COF based on 2,5-dihydroxyterephthalaldehyde and 1,3,5-tris(4-aminophenyl)benzene including a paper in Nature Communication in 2015. Authors modified this available COF by introducing sulfonation groups.

2, The procedure that introduced sulfonation groups also has some limitations as shown in Figure 2. Figure 2b points out that modified materials lose ordered structure and Figure 2d showed significant decrease of surface areas. One paper proved that the use of solvents like methanol make some pore collapse.⁴ Therefore, porous and high surface area, a general quality of COFs as written by authors, finally failed to obtain. Surface area reduced significantly by over 3 times from about 1300 m²/g of the first COFs to about 380 m²/g of the final modified materials. Despite the significant reduce in surface area, authors kept emphasizing the benefit of porous structure and referring this property as a reason for the obtained actuation performance.

3, Authors wrote in the manuscript: "...created by π - π interaction between the two-dimensional (2D) COF layers, may provide effective charge transfer pathways and improve capacities based on a pseudocapacitive mechanism...", which is an electrode property and is bad for electrolyte. Because when electrolyte can conduct electrons, to some extent, electrons can directly go from one electrode to another, which should be avoided. Otherwise, ion migrations could be reduced. Therefore, how do authors explain for the use of COFs in electrolyte here since the whole conjugated π system of your COF could conduct electrons, which eventually lead to short circuit?

At abstract, authors wrote that "...naturally ordered pore structure of the electron transport and ...". In introduction, authors wrote that "surface area give appealing benefits for increased electron and ion transport via the frameworks". However, in result, authors claimed that "we conducted the impedance measurement on the equivalent COF-DT ..., the results showed that both materials were electrical insulators". How do authors respond to these disagreements? Authors did not provide how you performed EIS. However, if EIS was measured at high frequencies and small amplitude of applied voltages much far from those of actuator operation, the electrical conductivity of materials in the two different conditions could be not similar. An insulator at very high frequency and low amplitude of applied voltages could be a conductor at very low frequency and high voltages.

4, Because COFs in electrolyte with whole conjugated π system could conduct electrons, the conductivity provide in Figure 3b could be a combined result of both electron conductivity and ionic conductivity because ionic conductivity was not measured directly, but indirectly via measuring resistance (Figure 3a). Therefore, authors assumed that the calculated conductivity is ionic conductivity and claimed this material to be a superionic conductor. This could be not correct unless more data should be provided to clearly prove the authors' idea.

5, Authors claimed that COF-DT-SO₃Na had higher ionic conductivity than COF-DT-SO₃H and COF-DT-SO₃H is an insulator, which could be not true. These two materials have the same structure except for different cations. The fact is that the size of Na⁺ is much bigger than that of H⁺ and the mobility of Na⁺ is much slower than that of H⁺. Therefore, COF-DT-SO₃Na should have lower ionic conductivity than COF-DT-SO₃H. I wonder if authors prepared samples and measured EIS at the same condition or not.

6, COFs usually exist in solid powders so do your materials as showed in Figure 1. Therefore, to

make membranes, they require polymer binder such as PEG6. The use of insulated polymer binder also prevents short circuit as mentioned in question 4. Pure solid powder COFs without any binders require very high pressure to form pellets or films⁷. Here, authors also prepared pellets for ionic conductivity measurement. However, for actuator fabrication, authors simply dispersed COFs in an organic solvent mixture at low concentration of 10mg/ml, then cast film directly on PEDOT:PSS film at 100oC for one hour. This fabrication could result in poor mechanical strength. Could authors provide data regarding to mechanical strength of your COF film/membrane without PEDOT:PSS and actuators (COF and PEDOT:PSS)?

7, Authors claimed: "we demonstrate an oriented ionic two-15 dimensional (2D) covalent-organic framework (COF)". However, authors provided no proofs that this "vertically oriented" structure was obtained by simply casting COFs dispersion on a PEDOT:PSS film. In nano scale within a single COF particle, structure is oriented. However, in macro scale, each particle could orient in different directions. Even with special treatments, making all particles oriented in the same direction with the pores vertically to PEDOT:PSS film could be very difficult. How do author explain about this?

8, By simply casting COFs powder dispersions on PEDOT:PSS film, does it form strong bonding between COFs and PEDOT:PSS? Authors only referred to cross-sectional SEM image without any value data and comparison to claim strong bonding between PEDOT:PSS electrode and COFs electrolyte, which is not proper and insufficient.

9, When making actuators, PEDOT:PSS was directly cast on a glass slide. The obtained PEDOT:PSS film can form strong bonding to glass. How did authors remove actuators from glass slides while remaining pristine PEDOT:PSS electrode?

10, Actuation performance is not quite good and response time is not correct. For example, although this actuator is very thin, about 40 μm , its bending is only 9.6 mm at 0.5V and 0.1 Hz. An actuator with the same length and much higher thickness of about 115 μm and made of COFs electrodes showed about double bending (17 mm) at the same excited conditions.⁸ Response time was calculated from AC excitation, which is not correct. Providing DC response could be more proper. Authors referred to strong resonance frequency as a good property. However, resonance usually need to be avoided because it could damage actuators. Furthermore, data got around resonance are not correct because the values much depend on actuator dimensions rather than depending on material properties. Blocking force is about 1.5 mN, which is low.

11, How do authors explain about the disagreement between Figure 3e and 4a? In Figure 3e, at 0.1Hz, displacement kept increasing to the maximum value, which means that at higher frequencies, displacements decrease. For example, from this figure, rough calculation results in displacement of over 6mm at 0.5Hz. However, Figure 4a showed that displacement unchanged when frequency increased from 0.1 to 1 Hz.

Provided data for both materials and actuators are very limited and insufficient to support authors' ideas. Many controversy and unclear issues remain. Major revision is needed.

Reference

- 1 Mo, Y. P. et al. The intramolecular H-bonding effect on the growth and stability of Schiff-base surface covalent organic frameworks. *Phys Chem Chem Phys* 19, 539-543, doi:10.1039/c6cp06894d (2017).
- 2 Zuo, H. Y., Li, Y. & Liao, Y. Z. Europium Ionic Liquid Grafted Covalent Organic Framework with Dual Luminescence Emissions as Sensitive and Selective Acetone Sensor. *Acs Appl Mater Inter* 11, 39201-39208, doi:10.1021/acsami.9b14795 (2019).
- 3 Qian, H. L., Dai, C., Yang, C. X. & Yan, X. P. High-Crystallinity Covalent Organic Framework with Dual Fluorescence Emissions and Its Ratiometric Sensing Application. *ACS Appl Mater Interfaces* 9, 24999-25005, doi:10.1021/acsami.7b08060 (2017).
- 4 Zhu, D. & Verduzco, R. Ultralow Surface Tension Solvents Enable Facile COF Activation with Reduced Pore Collapse. *ACS Appl Mater Interfaces* 12, 33121-33127, doi:10.1021/acsami.0c09173

(2020).

5 Kandambeth, S. et al. Self-templated chemically stable hollow spherical covalent organic framework. *Nat Commun* 6, 6786, doi:10.1038/ncomms7786 (2015).

6 Guo, Z. et al. Fast Ion Transport Pathway Provided by Polyethylene Glycol Confined in Covalent Organic Frameworks. *J Am Chem Soc* 141, 1923-1927, doi:10.1021/jacs.8b13551 (2019).

7 Montoro, C. et al. Ionic Conductivity and Potential Application for Fuel Cell of a Modified Imine-Based Covalent Organic Framework. *J Am Chem Soc* 139, 10079-10086, doi:10.1021/jacs.7b05182 (2017).

8 Mahato, M. et al. CTF-based soft touch actuator for playing electronic piano. *Nat Commun* 11, 5358, doi:10.1038/s41467-020-19180-3 (2020).

Reviewer #2 (Remarks to the Author):

Lee and co-workers reported in the manuscript a porous ionic electrolyte (COF-DT-SO₃Na) for electrochemical soft actuators with good actuation performance. The as-prepared actuators showed high response-ability and stability, and the soft actuator were demonstrated for potential practice application. Overall speaking, this work is interesting. However, in its current form, the manuscript suffers from a lack of clarity and some necessary experiments are missing. Revisions are needed to address the comments below:

1. The COF-DT-SO₃Na here acts as provider of nanoporous channels or as the electrolyte? Does the particle size of COF-DT-SO₃Na influence the transfer rate of ions? The size distribution of COF-DT-SO₃Na particle should be provided.
2. The pore size distribution of COF-DT-SO₃Na from 77K N₂ adsorption test should be provided.
3. The author should make a performance comparison of this ionic COF material and other state-of-art materials.
4. The temperature of casting the COF-DT-SO₃Na layer in methods is inconsistent with the figure 2(c), the author should correct it.
5. The citation format of ref 41 is incorrect.

REVIEWER COMMENTS

Reviewer #1:

Comment: This paper reports a use of COF as electrolyte in ionic actuators. The paper was well written and includes many interesting results. However, although authors claimed at the end of the second paragraph in the introduction that: “As far as we know, the potential of high-performance two-dimensional ionic COF-based electrolytes in artificial actuators has largely remained unexplored”, the research presented here has many critical and fundamental issues as pointed out following:

Response: Thanks for the reviewer contributing to our work and providing constructive feedback. All the significant issues have been carefully considered and addressed in the revised manuscript or the revised supplementary information.

1. The material is not totally new. Many papers 1-4 reported about the COF based on 2,5-dihydroxyterephthalaldehyde and 1,3,5-tris(4-aminophenyl)benzene including a paper in Nature Communication in 2015. Authors modified this available COF by introducing sulfonation groups.

Response: Thank you for your comments. COF has been under development for almost two decades. A wide range of COF materials has been synthesized and investigated. Although some references reported the COF based on 2,5-dihydroxyterephthalaldehyde and 1,3,5-tris(4-aminophenyl)benzene, however, no research has been conducted to investigate the usage of its derivatives as the electrolyte layer of electrochemical actuators. Here, we employ an ionic COF-DT-SO₃Na as the electrolyte material, which not only supply ions for the actuation performance but also contains matching pores to facilitate ion transmission, resulting in a large peak to peak displacement (9.3 mm, @±0.5 V, 1 Hz) which is higher than some other state-of-art electrochemical actuators (Supplementary table 4), a short time to reach equilibrium-bending motion (~1 s), a correspondingly high bending strain difference (0.38%, @±0.5 V, 1 Hz), a broad response frequency (0.1–20 Hz) and excellent durability (>99%) after 23,000 cycles.

2. The procedure that introduced sulfonation groups also has some limitations as shown in Figure 2. Figure 2b points out that modified materials lose ordered structure and Figure 2d showed significant decrease of surface areas. One paper proved that the use of solvents like methanol make some pore collapse.⁴ Therefore, porous and high surface area, a general quality

of COFs as written by authors, finally failed to obtain. Surface area reduced significantly by over 3 times from about 1300 m²/g of the first COFs to about 380 m²/g of the final modified materials. Despite the significant reduce in surface area, authors kept emphasizing the benefit of porous structure and referring this property as a reason for the obtained actuation performance.

*Response: Thank you for your comments, we apologize for the confusing way of plotting Fig. 2b, which caused the reviewer's confusion, we have replotted Fig. 2b to display the PXRD peaks clearly for 3 samples. According to the PXRD pattern, the modified COFs (COF-DT-SO₃H and COF-DT-SO₃Na) retain an ordered structure when the sulfonate group is introduced. The alkyl sulfonate chain occupies part of the empty space of COF-DT (3.2 nm) and makes the channel size narrower (2.4 nm), accounting for the significant decrease in surface areas. Despite the lower surface area, the pore size distribution (PSD) in supplementary Fig. 3 shows that the COF-DT-SO₃Na has a reasonable pore size (2.4 nm) for transporting ions. This is in agreement with a recent literature. (Sun, Q., et al. Reaction environment modification in covalent organic frameworks for catalytic performance enhancement. Angew. Chem. Int. Ed. **58**, 8670-8675 (2019). In contrast, although having a larger surface area, the unsulfonated sample (COF-DT) does not exhibit ionic conductivity and thus cannot function as an electrolyte for the actuators.*

3. Authors wrote in the manuscript: "...created by π - π interaction between the two-dimensional (2D) COF layers, may provide effective charge transfer pathways and improve capacities based on a pseudocapacitive mechanism...", which is an electrode property and is bad for electrolyte. Because when electrolyte can conduct electrons, to some extent, electrons can directly go from one electrode to another, which should be avoided. Otherwise, ion migrations could be reduced. Therefore, how do authors explain for the use of COFs in electrolyte here since the whole conjugated π system of your COF could conduct electrons, which eventually lead to short circuit?

Response: Thanks for the reviewer to point out the mistake. It should be "ion transport", instead of "electron transfer". It will provide effective ion transport pathways. To evaluate the potential electronic conductivity of COF-DT-SO₃Na, we examined the impedance of the geometrically equivalent COF-DT and COF-DT-SO₃H without adding the EC/DMC mixed solvent (25 °C, 30% RH). The results reveal that both materials were electric insulators. (Supplementary Fig. 4 and 5). To further evaluate the potential electronic conductivity of all the COF samples, we utilized a 4-probe probe station (Keithley 4200SCS) to conduct the resistance. The result demonstrates that all COF samples (COF-DT, COF-DT-SO₃H, COF-DT-SO₃Na) are electrical insulators ($< 1 \times 10^{-9} \text{ S cm}^{-1}$). The mistakes have been addressed in the

revised manuscript.

4. At abstract, authors wrote that "...naturally ordered pore structure of the electron transport and ...". In introduction, authors wrote that "surface area give appealing benefits for increased electron and ion transport via the frameworks". However, in result, authors claimed that "we conducted the impedance measurement on the equivalent COF-DT ..., the results showed that both materials were electrical insulators". How do authors respond to these disagreements? Authors did not provide how you performed EIS. However, if EIS was measured at high frequencies and small amplitude of applied voltages much far from those of actuator operation, the electrical conductivity of materials in the two different conditions could be not similar. An insulator at very high frequency and low amplitude of applied voltages could be a conductor at very low frequency and high voltages.

Response: Thanks for the reviewer in pointing out the mistake. This is the same concern as comment 3, it should be "ion transport" rather than "electron transport." These mistakes have been addressed in the revised manuscript. "Ionic conductivity measurements were performed on sample pellets using CHI 760E workstation over a frequency range from 1 MHz to 1 Hz and with an input voltage amplitude of 10 mV (30 % RH, 25 °C). The sample pellets were tightly connected between two platinum electrodes by means of spring, to ensure good contact between sample and each electrode." The details of EIS measurements were included in the supplementary information.

5. Because COFs in electrolyte with whole conjugated π system could conduct electrons, the conductivity provide in Figure 3b could be a combined result of both electron conductivity and ionic conductivity because ionic conductivity was not measured directly, but indirectly via measuring resistance (Figure 3a). Therefore, authors assumed that the calculated conductivity is ionic conductivity and claimed this material to be a superionic conductor. This could be not correct unless more data should be provided to clearly prove the authors' idea.

Response: Thanks for the comments. To evaluate the potential electronic conductivity of COF-DT-SO₃Na, we examined the impedance of the geometrically equivalent COF-DT and COF-DT-SO₃H without adding the EC/DMC mixed solvent (25 °C, 30% RH). The results reveal that both materials were electric insulators (Supplementary Fig. 4 and 5). To further measure electric conductivity, we utilized a 4-probe probe station (Keithley 4200SCS). The result demonstrates that all COF samples (COF-DT, COF-DT-SO₃H, COF-DT-SO₃Na) are electrical insulators ($< 1 \times 10^{-9} \text{ S cm}^{-1}$). Hence, the estimated conductivity of COF-DT-SO₃Na is ionic conductivity.

6. Authors claimed that COF-DT-SO₃Na had higher ionic conductivity than COF-DT-SO₃H and COF-DT-SO₃H is an insulator, which could be not true. These two materials have the same structure except for different cations. The fact is that the size of Na⁺ is much bigger than that of H⁺ and the mobility of Na⁺ is much slower than that of H⁺. Therefore, COF-DT-SO₃Na should have lower ionic conductivity than COF-DT-SO₃H. I wonder if authors prepared samples and measured EIS at the same condition or not.

Response: What we meant is that COF-DT-SO₃Na possesses ionic conductivity when EC/DMC mixed solvents are present, as shown in Figure 4a. In contrast, COF-DT and COF-DT-SO₃H show poor conductivity in the absence of solvent (Figure S4 and S5). To proof the conductivity in the same condition, ie. without presence of solvent, the electrical conductivity is almost absent for all samples as answered in Q5. We have rephrased the experimental conditions in the revised manuscript, page 4.

7. COFs usually exist in solid powders so do your materials as showed in Figure 1. Therefore, to make membranes, they require polymer binder such as PEG⁶. The use of insulated polymer binder also prevents short circuit as mentioned in question 4. Pure solid powder COFs without any binders require very high pressure to form pellets or films⁷. Here, authors also prepared pellets for ionic conductivity measurement. However, for actuator fabrication, authors simply dispersed COFs in an organic solvent mixture at low concentration of 10mg/ml, then cast film directly on PEDOT:PSS film at 100 °C for one hour. This fabrication could result in poor mechanical strength. Could authors provide data regarding to mechanical strength of your COF film/membrane without PEDOT:PSS and actuators (COF and PEDOT:PSS?)

Response: Thanks for the comments. The COFs are always present in powder form. The COF-DT-SO₃Na in our situation is an electrical insulator. A short circuit will not occur. The COF-DT-SO₃Na can be evenly dispersed in the NMP solvent. We attempted to make the COF-DT-SO₃Na film, however, we were unable to obtain a free-standing membrane. As a result, we cannot obtain the mechanical strength of COF-DT-SO₃Na film/membrane without PEDOT:PSS and actuators. We did not add any binder, such as PVDF, since we did not want to introduce any additional variables that would interfere with our understanding of COF performance. This is why we use PEDOT:PSS as the supporting (bottom) layer to fabricate the COF-DT-SO₃Na based electrochemical actuators.

8. Authors claimed: “we demonstrate an oriented ionic two-dimensional (2D) covalent-organic framework (COF)”. However, authors provided no proofs that this “vertically oriented”

structure was obtained by simply casting COFs dispersion on a PEDOT:PSS film. In nano scale within a single COF particle, structure is oriented. However, in macro scale, each particle could orient in different directions. Even with special treatments, making all particles oriented in the same direction with the pores vertically to PEDOT:PSS film could be very difficult. How do author explain about this?

Response: Thanks for the comment. We agree with the reviewer's comment; it is very difficult to control the macro scale orientation in the actuator device. What we are referring here, is the structural orientation of the COF that has proven using PXRD analyses. To minimize any confusion, we have removed the words "vertically" in the revised manuscript.

9. By simply casting COFs powder dispersions on PEDOT:PSS film, does it form strong bonding between COFs and PEDOT:PSS? Authors only referred to cross-sectional SEM image without any value data and comparison to claim strong bonding between PEDOT:PSS electrode and COFs electrolyte, which is not proper and insufficient.

Response: Thanks for the comments. In the previous submission, we wrote "strong interlayer adhesion", but not "strong bonding". Due to the COF-DT-SO₃Na and the PSS both have the sulfonate groups in the structures, the PEDOT will establish an interaction between the COF-DT-SO₃Na structures. This is one of the reasons we choose sulfonate COF-DT-SO₃Na as electrolyte. Long-term durability testing also demonstrates the actuator device is stable. To avoid any misunderstandings, we remove the word "strong" from the text. In contrast, we show a cross-sectional SEM image of COF-DT sample without interlayer adhesion.

(COF-DT device, without interlayer adhesion)

10. When making actuators, PEDOT:PSS was directly cast on a glass slide. The obtained PEDOT:PSS film can form strong bonding to glass. How did authors remove actuators from

glass slides while remaining pristine PEDOT:PSS electrode?

Response: Thanks for the comment. Although PEDOT:PSS adheres to the glass slide, it is readily detached and peeled by the blade.

11. Actuation performance is not quite good and response time is not correct. For example, although this actuator is very thin, about 40 μm , its bending is only 9.6 mm at 0.5V and 0.1 Hz. An actuator with the same length and much higher thickness of about 115 μm and made of COFs electrodes showed about double bending (17 mm) at the same excited conditions.⁸ Response time was calculated from AC excitation, which is not correct. Providing DC response could be more proper. Authors referred to strong resonance frequency as a good property. However, resonance usually need to be avoided because it could damage actuators. Furthermore, data got around resonance are not correct because the values much depend on actuator dimensions rather than depending on material properties. Blocking force is about 1.5 mN, which is low.

Response: Thanks for the comments. By comparing the displacement is not comprehensive. The displacement is determined by the size, thickness and the leaser point detachment of the actuator, as well as the voltage and the frequency applied. The bending strain difference (%) is more significant. It contains the thickness, the tip displacement, and the free length of the actuator. We made a table to compare the bending performance of the state-of-art ionic soft actuators (Supplementary Table 4). It demonstrates that COF-DT- SO_3Na actuator has competitive performance than other materials under ± 0.5 V, 1 Hz.

*Sorry for the irregular expression of response time, we were attempting to emphasize the amount of time it takes to reach equilibrium-bending motion (~ 1 s), which we've rephrased in the revised manuscript. The response time was described to be < 50 ms, when tested at > 20 Hz for the time taken to detect a motion, according to the reference (Ma, S., et al. High-Performance Ionic-Polymer-Metal Composite: Toward Large-Deformation Fast-Response Artificial Muscles. *Adv. Funct. Mater.* **30**, 1908508 (2020)). However, in another reference (Shi, Y., et al. Soft Electrochemical Actuators with a Two-Dimensional Conductive Metal-Organic Framework Nanowire Array. *J. Am. Chem. Soc.* **143**, 4017-4023 (2021)), they use the reach equilibrium-bending motion time as the response time, there seems to be no standard definition for the response time. To minimize confusion, we have changed it to reach equilibrium-bending motion time in the revised manuscript. We also provide the peak-to-peak displacement of the actuator under 0.5 V DC voltage (Supplementary Fig. 6).*

For the resonance of the actuators, it depends on how one uses it. There are numerous actuator examples that demonstrate the use of the resonance frequency. (1. Gao, X., et al.

Piezoelectric Actuators and Motors: Materials, Designs, and Applications. Adv. Mater. Technol. 5, 1900716 (2020); 2. Ji, X., et al. An autonomous untethered fast soft robotic insect driven by low-voltage dielectric elastomer actuators. Sci. Robot. 4, eaaz6451 (2019); 3. Mao, G., et al., Soft electromagnetic actuators. Sci. Adv. 6, eabc0251 (2020); 4. Zhao, H., et al. Compact Dielectric Elastomer Linear Actuators. Adv. Funct. Mater. 28, 1804328 (2018); 5. Ma, S., et al. High-Performance Ionic-Polymer-Metal Composite: Toward Large-Deformation Fast-Response Artificial Muscles. Adv. Funct. Mater. 30, 1908508 (2020).

It is true that the displacement data obtained around resonance frequency will not accurately represent the displacement of the actuator. In our example, it reveals the device can display the resonance movement under certain conditions. There are also many references reported the resonance of the actuators. (1. Maziz, A., et al. Demonstrating kHz Frequency Actuation for Conducting Polymer Microactuators. Adv. Funct. Mater. 24, 4851(2014); 2. Lu, L., et al. Graphene-Stabilized Silver Nanoparticle Electrochemical Electrode for Actuator Design. Adv. Mater. 25, 1270 (2013); 3. Roy, S. et al, Electroionic Antagonistic Muscles Based on Nitrogen-Doped Carbons Derived from Poly(Triazine-Triptycene). Adv. Sci. 4, 1700410 (2017); 4. Ma, S., et al. High-Performance Ionic-Polymer-Metal Composite: Toward Large-Deformation Fast-Response Artificial Muscles. Adv. Funct. Mater. 30, 1908508 (2020).)

For the blocking force, we made a table of the performance comparison (including blocking force) of this ionic COF material and other state-of-art materials (Supplementary Table 4).

12. How do authors explain about the disagreement between Figure 3e and 4a? In Figure 3e, at 0.1Hz, displacement kept increasing to the maximum value, which means that at higher frequencies, displacements decrease. For example, from this figure, rough calculation results in displacement of over 6 mm at 0.5 Hz. However, Figure 4a showed that displacement unchanged when frequency increased from 0.1 to 1 Hz.

Response: Thanks for the comments. The earlier data was acquired at the initial cycling before stabilization or equilibrium state, in order to be consistent, we have recollected the data as shown in Fig.4e, which shows consistent outcomes. Please see the revised manuscript for further information.

Provided data for both materials and actuators are very limited and insufficient to support authors' ideas. Many controversy and unclear issues remain. Major revision is needed.

Response: Thanks for the reviewer contributing to our work and providing constructive feedback. We have responded to all of the reviewer's comments. Please see the revised

manuscript for further information.

Reviewer #2:

Comment: Lee and co-workers reported in the manuscript a porous ionic electrolyte (COF-DT-SO₃Na) for electrochemical soft actuators with good actuation performance. The as-prepared actuators showed high response-ability and stability, and the soft actuator were demonstrated for potential practice application. Overall speaking, this work is interesting. However, in its current form, the manuscript suffers from a lack of clarity and some necessary experiments are missing. Revisions are needed to address the comments below:

Response: Thanks for the reviewer contributing to our work and providing constructive feedback.

1. The COF-DT-SO₃Na here acts as provider of nanoporous channels or as the electrolyte? Does the particle size of COF-DT-SO₃Na influence the transfer rate of ions? The size distribution of COF-DT-SO₃Na particle should be provided.

Response: Thanks for your comments, in this work, The COF-DT-SO₃Na will not only offer nanopore channels but will also function as an electrolyte layer. The particle size may have small influence to the transfer rate of ions. (see the figure below) (Strauss, F., Impact of cathode material particle size on the capacity of bulk-type all-solid-state batteries. ACS Energy Lett., 3, 992-996 (2018)). In our work, we are trying to make the particle size smaller (~ 200 nm), resulting in a uniform film, thus, we can obtain a stable actuator. The particle size of COF-DT-SO₃Na was ~ 200 nm after ultrasonic treatment. The size distribution of COF-DT-SO₃Na were added in the supplementary Fig. 8.

Figure 4. Mean ionic and electronic partial conductivities of ASSB cathode composites using NCM-L, NCM-M, and NCM-S.

2. The pore size distribution of COF-DT-SO₃Na from 77K N₂ adsorption test should be provided.

Response: Thanks for the comment, the pore size distribution (PSD) of COF-DT, COF-DT-

SO₃H, and *COF-DT-SO₃Na* have been included in supplementary Fig. 3. Please see the revised manuscript for further information.

3. The author should make a performance comparison of this ionic COF material and other state-of-art materials.

Response: Thanks for your good suggestion, we made a table of the performance comparison of this ionic COF material and other state-of-art materials (Supplementary Table 4).

4. The temperature of casting the *COF-DT-SO₃Na* layer in methods is inconsistent with the figure 3(c), the author should correct it.

Response: Thank you for pointing out the mistake; it has been corrected in the revised manuscript.

5. The citation format of ref 41 is incorrect.

Response: Thanks for your comment, it has been corrected in the revised manuscript.

REVIEWER COMMENTS

Reviewer #1 (Remarks to the Author):

The revised manuscript is now acceptable.

Reviewer #2 (Remarks to the Author):

The authors have satisfactorily addressed all the comments from the reviewer and it now can be accepted as is.

REVIEWERS' COMMENTS

Reviewer #1 (Remarks to the Author):

The revised manuscript is now acceptable.

Response: Thanks for the reviewer contributing to our work.

Reviewer #2 (Remarks to the Author):

The authors have satisfactorily addressed all the comments from the reviewer and it now can be accepted as is.

Response: Thanks for the reviewer contributing to our work.